# Trajectory-wise Iterative Reinforcement Learning Framework for Auto-bidding

## ABSTRACT

In online advertising, advertisers participate in ad auctions to acquire ad opportunities, often by utilizing auto-bidding tools provided by demand-side platforms (DSPs). State-of-the-art auto-bidding algorithms typically employ reinforcement learning (RL). However, due to safety concerns, most current RL-based auto-bidding policies are trained in simulated systems, leading to a performance degradation when deployed in online environments. To narrow this gap, we can deploy multiple auto-bidding agents to run in parallel, thereby collecting a large interaction dataset. Offline RL algorithms can then be utilized to train a new policy. The trained policy can subsequently be deployed for further data collection, resulting in an iterative training framework, which we refer to as iterative offline RL. In this work, we identify the performance bottleneck of this iterative offline RL framework, which originates from the ineffective exploration and exploitation caused by the inherent conservatism of offline RL algorithms. To overcome this bottleneck, we propose Trajectory-wise Exploration and Exploitation (TEE), which introduces a novel data collecting and data utilization method for iterative offline RL from a trajectory perspective. Furthermore, to ensure the safety of online exploration while preserving the dataset quality for TEE, we propose Safe Exploration by Adaptive Action Selection (SEAS). Both offline experiments and real-world experiments on Alibaba display advertising platform demonstrate the effectiveness of our proposed method.

## 1 INTRODUCTION

Online advertising [7] is becoming one of the major sources of profit for Internet companies. Due to the complex online advertising environments, auto-bidding tools provided by demand-side platforms (DSPs) are commonly utilized to bid on behalf of advertisers to optimize their advertising performance. Bidding for arriving ad impressions can be viewed as a sequential decision-making problem, and thus state-of-the-art auto-bidding algorithms leverage reinforcement learning (RL) to optimize bidding policies [4, 13, 32].

However, due to safety concerns, current RL-based auto-bidding policies are trained in simulated environments. Policies trained with simulation are shown suboptimal when deployed in the real-world system [22]. Therefore, it is desirable to optimize the bidding policy by directly interacting with the online environments. Classic (online) RL algorithms alternate frequently between data collection and policy update, and typically require enormous samples (i.e., transition tuples) to achieve convergence. However, collecting transition data can be extremely time-consuming, e.g., in most RL formulations of the auto-bidding problem [13, 32], an RL episode corresponds to 24 hours, and thus training a policy may take a long time. Additionally, frequent updates of online policies may cause unstable performance and potentially risky bidding behaviours.

To address these issues, DSPs usually leverage a large number of auto-bidding agents as parallel workers to allow for the efficient collection of large amount of interaction data, and train an auto-bidding policy on the dataset with offline RL [11, 19] — a data-driven RL paradigm that aim to extract policies from large-scale pre-collected datasets. The updated policy could again be deployed for further data collection, which results in an iterative framework for data collection and policy update. We refer to the above training paradigm as iterative offline RL, as depicted in Figure 1. Iterative offline RL presents a promising solution for online policy training in industrial scenarios, and similar approaches have also been mentioned in several academic works [21, 22, 36].

To facilitate effective policy improvement for each iteration in iterative offline RL, it is crucial for the collected datasets to encompass sufficient information regarding various states and actions. This often requires constructing an exploration policy that incorporates a degree of randomness, typically achieved by introducing noise perturbation to actions [20–22]. Nonetheless, employing a random exploration policy can adversely affect the performance of the trained policy. This is because the introduced randomness undermines the exploration policy's performance, and offline RL algorithms heavily relies on the exploration policy due to the principle of conservatism (or pessimism) [10, 11, 16, 18, 28], which compels the newly updated policy to be close to the exploration policy. Even if we have collected a dataset with rich information about the advertising environment, offline RL algorithms may not fully exploit its potential due to the influence of low-quality actions, resulting in the failure to learn a good policy. We demonstrate this phenomenon in Section 4, highlighting the challenges of effective exploration and exploitation for iterative offline RL.

In this work, we tackle the aforementioned challenges by adopting a trajectory perspective for both the exploration (data collection) process and the exploitation (offline RL training) process. For efficient exploration, we construct exploration policies by introducing noise into the policy's parameter space instead of the traditional action space. This choice is motivated by our key observation that this injection of parameter space noise (PSN) yields an exploration dataset with a more dispersed trajectory return distribution (please refer to Section 5.1 for details). This observation indicates that the dataset contains a considerable number of high-return trajectories, which are valuable for offline training. For effective exploitation, we propose Robust Trajectory Weighting to fully exploit high-return trajectories in the collected dataset. Specifically, instead of uniformly sampling the dataset during training, we assign high probability weights to high-return trajectories, thereby enhancing the impact of high-quality behaviours on the training process and overcoming the conservatism problem. However, the instability of the advertising environments leads to highly stochastic trajectory returns that often fail to reliably reflect the trajectory qualities. To address this issue, we design a new trajectory quality indicator by

approximating the expectation of the stochastic rewards, in order to eliminate the effect of stochasticity and enhance the robustness of the trajectory weighting method. We leverage PSN for online exploration and utilize Robust Trajectory Weighting to compute sampling probabilities before offline RL training, boosting the effectiveness of exploration and exploitation in iterative offline RL.

Apart from effectiveness, safety constraints must also be taken into consideration during online exploration in real-world advertising systems. Random exploration can lead to risky bidding behaviours, negatively impacting the performance of advertisers. The safety of an exploration policy is captured by a performance lower bound, which ensures that the performance drop brought by exploration is acceptable. Ensuring safe exploration often requires imposing constraints on the original exploration policy. However, existing safety-guaranteeing methods [22] often lack awareness of action qualities. While they prevent dangerous behaviours, they also restrict some high-quality actions. This may hinder the emergence of high-return trajectories during exploration, and subsequently affect the performance of the training process. To preserve data quality while ensuring safety, we propose SEAS, which dynamically determines the safe exploration action at each time step based on the cumulative rewards up to that step and the predicted future return. By making adaptive decisions, SEAS preserves the quality of the collected dataset to the fullest extent, and achieves theoretically guaranteed safety at the same time.

Main contributions of this work are summarized as follows:

- We identify and demonstrate that the performance bottleneck of the current iterative offline RL paradigm for auto-bidding algorithms mainly lies in ineffective exploration and exploitation caused by the conservatism principle of offline RL algorithms.
- We propose TEE, a solution for effective exploration and exploitation in iterative offline RL for auto-bidding. TEE comprises two components: PSN for trajectory-wise exploration, and a novel Robust Trajectory Weighting algorithm for trajectory-wise exploitation.
- For safe exploration in auto-bidding, we design SEAS, which adaptively decides the safe exploration action for each time step based on the cumulative rewards till that step. SEAS enjoys provable safety guarantee while sacrificing minimal performance in policy learning when functioning together with TEE.
- Extensive experiments in both simulated environments and Alibaba display advertising platform demonstrate the effectiveness of our solution in terms of trained policy's performance, as well as the safety of the training process.

## 2 RELATED WORK

*Reinforcement learning for auto-bidding.* Bid optimization in online advertising is a sequential decision procedure, and can be solved via reinforcement learning techniques. Cai et. al. [4] first formulated the auto-bidding problem as an MDP. Wu et. al. [32] and He et. al. [13] leveraged reinforcement learning to optimize bidding policies under various constraints. All of the above works train their RL bidding agent in a simulated environment. Mou et. al. [22] recognized the

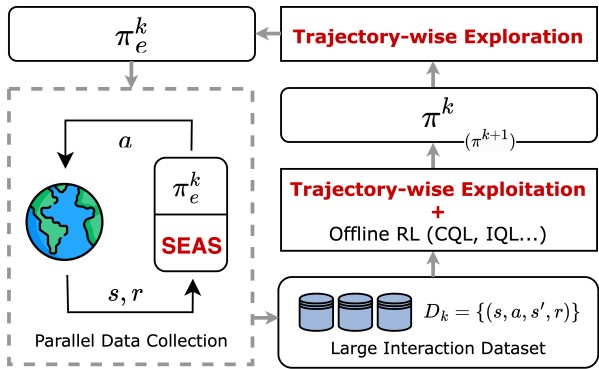

**Figure 1: Iterative offline RL with TEE and SEAS. Components proposed in this work are highlighted in red.**

sim2real problem in auto-bidding, and designed an iterative offline RL framework to train the bidding policy online.

*Offline RL.* Offline RL [11, 19, 27] (or batch RL) refers to the problem of policy optimization utilizing only previously collected data, without additional online interaction. Due to the distribution shift [11] problem that arises in offline RL, most algorithms conduct conservative policy learning, which compels the learning policy to stay close to the dataset. Various algorithms achieve this by directly regularizing the actor [10, 11, 17, 25], learning conservative value functions [18], or constraining the number of policy improvement steps [3, 16]. However, the conservatism principle causes the performance of trained policy to highly depend on the dataset quality.

*Dataset coverage in offline RL.* Dataset coverage is a core factor that limits the performance of offline RL algorithms. Schweighofer et. al. [29] conducted extensive experiments to demonstrate that dataset coverage is essential for offline RL algorithms to learn a good policy. Prior theoretical works on offline RL [5, 28, 35] also relied on datasets with sufficient state-action space coverage, which is often characterized by concentrability coefficients, to produce a strong performance guarantee.

*Exploration in RL.* Exploration is a crucial aspect of RL as it allows the agent to gather information about the environment. Traditional exploration strategies induce novel behaviours by random perturbations of actions, such as $\epsilon$-greedy [30] and entropy regularization [31]. To generate meaningful behavioural patterns for hard exploration tasks, several other approaches such as intrinsic reward-based exploration [6, 24], count-based exploration [2, 23] and PSN [9, 26] have been proposed. However, most of the existing solutions have been proposed for the online RL paradigm, which is substantially different from iterative offline RL where data is collected for training offline RL algorithms.

## 3 PRELIMINARIES

In this work, we consider auto-bidding with budget constraint, a sequential decision problem, where an advertiser submits bids for incoming ad impressions, aiming to maximize the total value within a fixed budget. For this problem, previous works [37, 38] showed that under the second price auction [8], the optimal bid $b^*$ on an

impression is given by $b^* = v/\lambda$, where $v$ represents the impression value (*e.g.* click-through rate) and $\lambda$ is a scaling factor. However, determining the optimal value of $\lambda$ in real time is intractable due to its dependence on values and costs of all impressions in the stream. Thus, we formulate the problem of adjusting bidding parameter $\lambda$ as a Markov Decision Process (MDP), defined by a tuple $(\mathcal{S}, \mathcal{A}, r, p, \gamma)$. In our formulation, an episode corresponds to a one-day ad campaign duration, which is divided into $T$ time steps. At each step $t$, the advertiser observes state $s_t \in \mathcal{S}$ and takes an action $a_t \in \mathcal{A}$. The state $s_t$ is a feature vector describing the advertising status of a campaign, which may contain time, remaining budget, budget consumption speed, and etc. The action $a_t$ is the bidding parameter $\lambda$ in time step $t$. The reward $r(s_t, a_t)$ is the total value of impressions won between time step $t$ and $t + 1$, and $p(s_{t+1}|s_t, a_t)$ denotes the transition probability of states. Both $r$ and $p$ are determined by the advertising environment. The discount factor $\gamma \in [0, 1]$ accounts for the future rewards' diminishing impact. A (deterministic) policy $\pi \in \Pi : \mathcal{S} \rightarrow \mathcal{A}$ is a function defining the agent's bidding behaviour. When a policy interacts with the advertising environment over an episode, a trajectory $\tau = \{(s_t, a_t, s_{t+1}, r_t)\}_{t=0}^{T}$ is generated[1], where the initial state $s_0$ is drawn from a probability distribution $\rho$.

The (discounted) return of trajectory $\tau$ is $R(\tau) = \sum_{t=0}^{T} \gamma^t r_t$. The objective of RL is to maximize the expected return:

$$\arg\max_{\pi \in \Pi} J(\pi) := \mathbb{E}_\tau[R(\tau)].$$

In RL, the state value function of a policy $\pi$ is defined as

$$V^\pi(s) := \mathbb{E}_\pi\Big[\sum_{u=t}^{T} \gamma^u r_u | s_t = s\Big], \quad \forall s \in \mathcal{S}.$$

Similarly, the action value function is

$$Q^\pi(s, a) := \mathbb{E}_\pi\Big[\sum_{u=t}^{T} \gamma^u r_u | s_t = s, a_t = a\Big], \quad \forall s \in \mathcal{S}, a \in \mathcal{A}.$$

Iterative offline RL follows a cyclical pattern of data collection and offline policy update, repeatedly for a total of $K$ iterations. Within each iteration $k \in [K]$, an exploration policy $\pi_e^k$ is constructed based on $\pi^k$. Then $\pi_e^k$ is deployed in the advertising environment to collect interaction dataset $D^k = \{\tau_i\}_{i=1}^N$ containing $N$ trajectories. An offline RL algorithm is then used to learn a policy from $D^k$, producing the updated policy $\pi^{k+1}$ for the subsequent iteration.

**Safety of Bidding Policies.** The performance of auto-bidding policies deployed in the advertising system must be guaranteed in either the stages of policy deployment or policy training. Therefore, safety of the RL training process is formally defined by a performance lower bound:

$$J(\pi_e^k) \geq (1 - \epsilon)J_s, \forall k = 1, \cdots, K,$$

where $J_s$ is the performance of a known safe policy. Note that the exploration policy $\pi_e^k$ in every iteration $k$ should be ensured to be safe. To satisfy the safety constraint in the first iteration, the training process should be initialized by a known safe (but could be

---
[1]For simplicity, we assume all trajectories' lengths are equal to episode length $T$, though a campaign may exhaust its budget at certain time $t_0 < T$ and cannot afford any impression thereafter. In such cases, we let $r(s_t, a_t) = 0, \forall t_0 \leq t \leq T$.

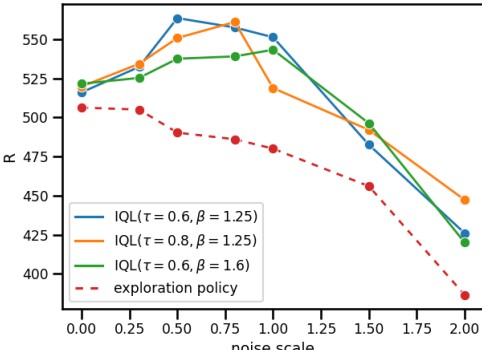

**Figure 2: Performance of exploration policies with different noise scale, as well as the performance of policies trained with IQL on datasets collected by those exploration policies.**

suboptimal) policy instead of a random policy. In practice, policies trained in a simulation [13] could serve as an initial policy.

## 4 PERFORMANCE BOTTLENECK OF ITERATIVE OFFLINE RL

In this section, we present empirical observations regarding the performance bottleneck of iterative offline RL, and also introduce the idea of our proposed method. In each iteration of iterative offline RL, the data-collection policy plays a crucial role in determining the input dataset for the subsequent offline training process, which in turn influences the trained policy. The data-collection process should gather sufficient information of the underlying MDP through effective exploration.

A conventional approach for exploration in iterative offline RL [21] is adding noise perturbations to actions. Concretely, an exploration policy with action space noise (ASN) is constructed as $\pi_e^{ASN}(s_t) = \pi(s_t) + \epsilon_t$, where $\epsilon_t \sim \mathcal{N}(0, \sigma^2 I)$ is sampled from a Gaussian distribution with standard deviation $\sigma$. The addition of noise aims to achieve sufficient coverage of the state-action space. However, a highly random policy often leads to suboptimal performance and results in a dataset consisting primarily of low-quality actions. Consequently, such datasets pose challenges for offline RL algorithms to produce high-quality policies in the subsequent training process. This is because the conservatism principle of these algorithms drives the learned policy close to the low-performing exploration policy.

We conduct experiments in a simulated environment (details provided in Appendix C) to illustrate the aforementioned issue. Based on one deterministic policy, we construct exploration policies by adding different levels of noise perturbation on the actions. These exploration policies are deployed in the environment to collect interaction datasets, which are then utilized by IQL [16], an offline RL algorithm, to train new policies. Figure 2 presents the performances of exploration policies with varying noise scale and performances of the trained policies. We can observe that adding a certain amount of noise boosts the trained policy by gaining more environment information, while an excessively noisy exploration policy undermines the training result. Furthermore, the optimal

noise scale depends on the training algorithm, making online tuning of the noise scale impractical.

One potential method for overcoming the difficulty brought by conservatism is to manually identify high-quality behaviours from the noisy dataset and allow the learning algorithm to focus solely on these behaviours. However, evaluating the quality of individual actions within a dataset can be challenging. For one transition tuple $(s, a, r, s')$, a large reward $r$ does not necessarily indicate $a$ to be a good action, due to the influence of $s'$ on future rewards. Hopefully, if a full trajectory $\tau$ attains a high return, then it is reasonable to infer that this trajectory contains high-quality behaviours. In following sections, we present empirical findings to reveal that employing PSN instead of ASN for exploration leads to a dataset containing more high-return trajectories. Based on this observation, we propose TEE for online RL in auto-bidding.

## 5 PROPOSED FRAMEWORK

In this section, we present our novel design on iterative offline RL framework for auto-bidding, which comprises two key components: TEE and SEAS. TEE is proposed to optimize the effectiveness of exploration (i.e., data collection) and exploitation (i.e., offline training), and SEAS is specifically developed to address the safety concerns that may arise during the online exploration process.

### 5.1 Trajectory-wise Exploration

We introduce Parameter Space Noise (PSN) [9, 26] for exploration in the iterative offline RL framework, and explain its effectiveness from a trajectory view. PSN refers to injecting noise in an RL policy's parameter space in order to induce exploratory behaviours. It has brought performance gain on a wide range of control tasks when applied to online deep RL algorithms [1, 14]. For a parameterized policy (e.g. a neural network) $\pi(s; \theta)$, where $\theta$ is the parameter vector, applying additive Gaussian noise to $\theta$ gives $\hat{\theta} = \theta + \epsilon$, where $\epsilon \sim \mathcal{N}(0, \sigma^2 I)$. Then the exploration policy based on PSN is $\pi_e^{PSN}(s_t) = \pi(s_t; \hat{\theta})$. Importantly, the perturbed parameter vector $\hat{\theta}$ is only sampled at the beginning of each episode and remains fixed afterwards. This is substantially different from ASN where independent noise is added at every time step.

We now present our key observation on datasets collected by PSN by experiments in a simulated bidding environment (details are provided in Appendix C). We first construct two exploration policies with ASN and PSN based on one policy $\pi$, and denote them by $\pi_e^{ASN}$ and $\pi_e^{PSN}$ respectively. Subsequently, two datasets $D^{ASN}$ and $D^{PSN}$ of equal size are collected with $\pi_e^{ASN}$ and $\pi_e^{PSN}$. To ensure a fair comparison, we control the noise strength of both exploration policies to guarantee that the average return of $D^{ASN}$ and $D^{PSN}$ are equal. As shown in Figure 3, the return distribution of trajectories in $D^{PSN}$ is more dispersed than that of $D^{ASN}$. Specifically, $D^{PSN}$ contains high-return trajectories (e.g. trajectories with return higher than 750) which are almost absent in $D^{ASN}$. Additionally, Table 1 shows that the return variance of $D^{PSN}$ is consistently higher than that of $D^{ASN}$ under different exploration degrees, indicating the potential of extracting desirable bidding behaviours from datasets collected by PSN policies.

The high return variance observed in PSN can be attributed to two main factors: decoupling between the exploration policy and

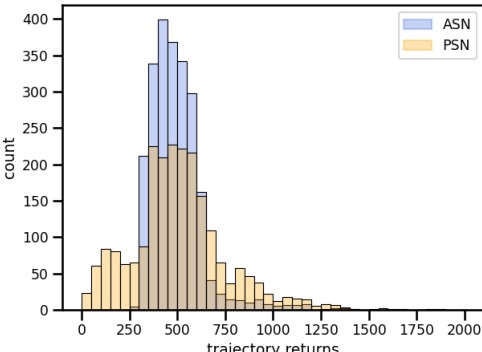

**Figure 3: Comparison of trajectory return distributions of datasets collected by ASN and PSN.**

**Table 1: Comparison of trajectory return variances of datasets collected by ASN and PSN.**

| Average Return | 400 | 450 | 500 | 550 |
|---|---|---|---|---|
| Return Variance of $D^{ASN}$ | 88.29 | 100.74 | 140.99 | 197.55 |
| Return Variance of $D^{PSN}$ | 236.07 | 246.10 | 245.40 | 265.46 |

the base policy (i.e. the original noiseless policy $\pi$), and the consistency of exploration behaviours. The following example provides further explanation in the context of auto-bidding, by highlighting the incapability of ASN on generating high-return trajectories. Assume that the current policy $\pi$ is suboptimal in that it produces relatively low bids across most states. When conducting exploration based on $\pi$, attaining high-return trajectories would require suggesting higher bids. In this case, positive perturbations $\epsilon_t$ are desirable when applying ASN. However, when $\pi_e^{ASN}(s_t)$ is lifted, $\pi(s_{t+1})$ would become low since the base policy $\pi$ aims to maintain the smoothness of budget consumption, thereby offsetting the effect of high bid in step $t$. The counterbalancing actions of the base policy substantially prevent ASN from effectively exploring unknown areas. Another issue with ASN is that, since the perturbations $\epsilon_t$ of different time steps are drawn from independent probability distributions, consistently realizing positive $\epsilon_t$ throughout an entire episode is barely possible. PSN does not suffer from those issues, since once the exploration policy is sampled, its behaviours are not interfered by the base policy.

PSN enables us to sample a variety of policy parameters, which could be distributed to a great number of advertising campaigns in a DSP to explore different bidding behaviours. These campaigns run in parallel and collect a large dataset covering a wide range of behaviours, thus achieving large-scale trajectory-wise exploration.

### 5.2 Trajectory-wise Exploitation

While PSN-based exploration policies could effectively generate valuable high-return trajectories, their collected datasets still consist primarily of inferior trajectories, as depicted in Figure 3. Therefore, the trained policy's performance is still limited by the inherent conservatism of offline RL. To alleviate this problem and fully exploit

the dataset, we propose Robust Trajectory Weighting for trajectory-wise exploitation.

We consider a dataset containing multiple trajectories $D = \{\tau_i\}_{i=1}^N$, where $\tau_i = \{(s_{i,t}, a_{i,t}, s_{i,t+1}, r_{i,t})\}_{t=0}^T$. Inspired by [15, 33, 34], instead of uniform sampling, we assign large sample probabilities to well-performing trajectories during training. A straightforward way to realize this is to assign weight $w_i$ for trajectory $\tau_i$ based on its return $R_i := R(\tau_i)$, as high return typically indicates good behaviours. Nonetheless, two main issues arise when applying return-based trajectory weighting in the auto-bidding task. Firstly, due to the instability of auction environments and user feedbacks, the reward function $r(s_t, a_t)$ is highly stochastic, thus a high return might not necessarily be achieved by a good policy, but rather a "lucky" trial that happens to obtain high rewards in most time steps. Secondly, trajectories within the dataset may come from different advertising campaigns, and the intrinsic characteristics of a campaign, such as its budget level and item category, also affect the trajectory return. Thus, directly comparing trajectories from different campaigns by their returns is unfair.

We address the first issue by learning a reward model $\bar{r}$ on the dataset for predicting the expected reward of a state-action pair:

$$\bar{r} = \arg\min_r \sum_{i=1}^N \sum_{t=0}^T (r(s_{i,t}, a_{i,t}) - r_{i,t})^2.$$

The reward model could be implemented by a neural network with the above loss function. Then the original stochastic rewards $r_{i,t}$ are replaced with their expectations $\bar{r}_{i,t} = \bar{r}(s_{i,t}, a_{i,t})$ to calculate $\bar{R}_i = \sum_{t=0}^T \gamma^t \bar{r}_{i,t}$, producing more robust quality indicators.

To deal with the second issue mentioned above, we regularize the trajectory returns by subtracting the value of the initial state of the trajectory, estimated as $\hat{V} = \arg\min_V \sum_{i=1}^N (V(s_{i,0}) - \bar{R}_i)^2$. The initial state typically contains information (e.g. the total budget) of the advertising campaign, therefore $\hat{V}(s_{i,0})$ provides estimation of the expected return of the campaign behind trajectory $i$. Our final indicator of trajectory quality is $\hat{A}_i$:

$$\hat{A}_i = (\bar{R}_i - \hat{V}(s_{i,0}))/\hat{V}(s_{i,0}), \quad \forall 1 \le i \le N.$$

The sample probability $w_{i,t}$ of transition tuple $(s_{i,t}, a_{i,t}, s_{i,t+1}, r_{i,t})$ is computed according to $\hat{A}_i$ as follows:

$$w_{i,t} \propto \exp(\hat{A}_i/\alpha), \quad \forall 1 \le i \le N, 0 \le t \le T,$$

where $\alpha \in \mathbb{R}^+$ is a temperature parameter, and the weights should be normalized to ensure $\sum_{i=1}^N \sum_{t=0}^T w_{i,t} = 1$.

After computing weights for dataset $D$, we could run any model-free offline RL algorithm (e.g. CQL, IQL) using the reweighted data sampling strategy. We provide a theoretical justification of Robust Trajectory Weighting in Appendix A, showing how it alleviates the problem brought by conservative algorithms.

## 5.3 Safe Exploration

Although TEE boosts the effectiveness of iterative offline RL, the safety of data-collecting policies, which is of great importance when training in real-world advertising systems, have not been considered. In this section, we propose a novel algorithm named SEAS to guarantee the safety of online exploration.

---

**Algorithm 1** Safe Exploration by Adaptive Action Selection (SEAS)

1: **Input:** Exploration policy $\pi_e$, $n$ distinct safe policies $\{\pi_s^i\}_{i=1}^n$ and their state-action value function $\{Q^{\pi_s^i}\}_{i=1}^n$, safe performance $J_s$ and safety coefficient $\epsilon \in (0, 1)$
2: **Initialize:** Sample initial state $s_0 \sim \rho(s)$, $temp \leftarrow 1$
3: **for** $t = 0, 1, \cdots, T$ **do**
4:    $a_e \leftarrow \pi_e(s_t), a_s \leftarrow \pi_s^{temp}(s_t)$
5:    $temp \leftarrow \arg\max_i Q^{\pi_s^i}(s_t, a_e), Q_{max} \leftarrow Q^{\pi_s^{temp}}(s_t, a_e)$
6:    **if** $\sum_{u=0}^{t-1} r_u + Q_{max} \ge (1 - \epsilon)J_s$ **then**
7:        $a_t \leftarrow a_e$
8:    **else**
9:        $a_t \leftarrow a_s$
10:   **end if**
11:   Take action $a_t$, observe $r_t, s_{t+1}$
12: **end for**

---

On the problem of safe exploration in auto-bidding, Mou et. al. [22] designed a method based on safety zone, to restrict the exploratory actions around a safe policy. Specifically, $\pi_e(s_t) \leftarrow clip(\pi_e(s_t), \pi_s(s_t) - \xi, \pi_s(s_t) + \xi)$, where $\pi_s$ is a known safe policy. Though this method is provably safe under some assumptions on the MDP, its safety heavily relies on the radius $\xi$ which is intractable to determine. Besides, this approach lacks awareness of the quality of original exploration actions, and poses constraint on both bad and good actions, thus hurting the quality of collected datasets. SEAS mitigates these problems through an adaptive design. The aim of SEAS is to prevent the low-performing trajectories caused by the original exploration policy (e.g. $\pi_e^{PSN}$) to emerge, while preserving the high-quality ones to the fullest extent.

The procedure of SEAS interacting with the environment for one episode is shown in Algorithm 1. In each step, SEAS selects between an exploratory action $a_e$ and a safe action $a_s$ according to the condition in line 6. Note that multiple safe policies are provided to the algorithm, and in each step the safe policy with maximum Q value is chosen for constructing the condition. Utilizing multiple safe policies instead of one loosens the restriction in line 6, thus better preserving exploratory actions of $\pi_e$. The algorithm is "adaptive" in the sense that its action in each step $t$ depends on the rewards accumulated till time $t$.

The following theorem shows that SEAS theoretically guarantees safety for any exploration policy $\pi_e$. Proof of Theorem 1 is provided in Appendix B.

THEOREM 1. *For any policy $\pi_e$ and any $\epsilon \in (0, 1)$, given safe policies $\{\pi_s^i\}_{i=1}^n$ that satisfy $J(\pi_s^i) \ge J_s, \forall 1 \le i \le n$, the expected return $\mathbb{E}_\tau[R(\tau)]$ of trajectories generated by SEAS satisfies $\mathbb{E}_\tau[R(\tau)] \ge (1 - \epsilon)J_s$.*

The advantages of SEAS are summarized as follows: (i) SEAS needs only one hyperparameter $\epsilon$, which is in the definition of safety and is straightforward to set. (ii) The safety of SEAS is provable without additional assumptions on the underlying MDP. (iii) Experiments demonstrate that when functioning together with TEE, SEAS exhibits minimal performance sacrifice compared to baseline methods.

## 5.4 Overall Framework

TEE and SEAS can be combined to form an iterative framework for online policy training in auto-bidding. The training process is initialized with the current policy running in the bidding system, which is typically suboptimal but safe. In each iteration $k$, we employ PSN for exploration based on the current policy $\pi^k$. We pick a large number of advertising campaigns, and independently sample parameter vectors for different campaigns' policies. The exploration policies are input to SEAS to guarantee safety. A subset of previous policies $\{\pi^\kappa\}_{\kappa=1}^{k-1}$ can be utilized as safe policies for SEAS, and $\epsilon$ is a pre-defined constant parameter through iterations. The Q functions of safe policies for SEAS could be obtained through different approaches. For policies trained by value-based or actor-critic RL algorithms, we can directly query the existing value network. Alternatively, new value networks can be fitted on the collected datasets. Specifically, for approximating the Q function of $\pi^\kappa$, we can perform SARSA-style policy evaluation on dataset $D^\kappa$. Although this yields underestimated values due to exploration when collecting $D^\kappa$, it does not incur violation of the safety constraint. This is because Theorem 1 still holds even when the input Q functions have underestimated values. The safe exploration policies are then distributed and deployed to multiple advertising campaigns, running in parallel for a specific duration (e.g. one day) to collect an interaction dataset $D^k$. Subsequently, we employ an offline RL algorithm (e.g. IQL) to perform policy training on $D^k$, where the sampling probabilities are computed using Robust Trajectory Weighting. The resulting trained policy $\pi^{k+1}$ becomes the input for the subsequent iteration, and the process continues iteratively.

## 6 EXPERIMENTS

We provide empirical evidence for the effectiveness of our approach by both simulated experiments and real-world experiments on Alibaba display advertising platform.

## 6.1 Overall Performance in a Simulated Environment

A simulated advertising system is constructed for all the offline experiments in this work. Details of the setup and hyperparameters are provided in Appendix C. The implementation code is available to ease reproducibility [2].

We test the effectiveness of TEE and SEAS by combining them with three different offline RL algorithms. We also implement three baselines: iterative versions of IQL with and without exploration, as well as SORL [22]. An expert policy is trained with TD3 [12], an online RL algorithm, to serve as a performance upper bound. Methods for comparison are listed below.

- **TEE+SEAS+IQL [16] / CQL [18] / TD3BC [10].** Iterative offline RL with the proposed TEE and SEAS, using IQL / CQL / TD3BC as the offline RL algorithm.
- **IterIQL+ASN.** Iterative offline RL using IQL as the offline RL algorithm. IQL is selected as a representative because it is the state-of-the-art model-free offline RL algorithm. Exploration policy $\pi_e^k$ is constructed by adding ASN on $\pi^k$.

[2]https://anonymous.4open.science/r/TEE/

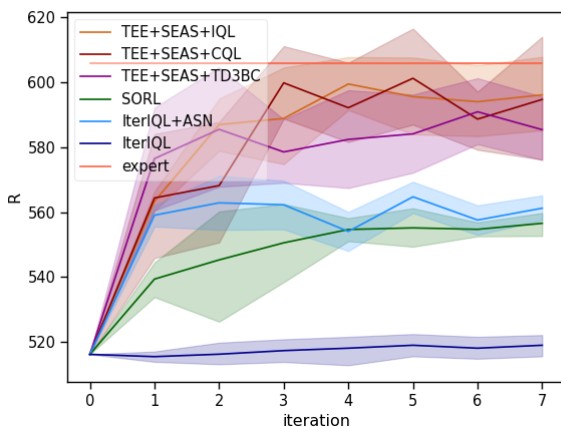

Figure 4: Overall performance in a simulated environment.

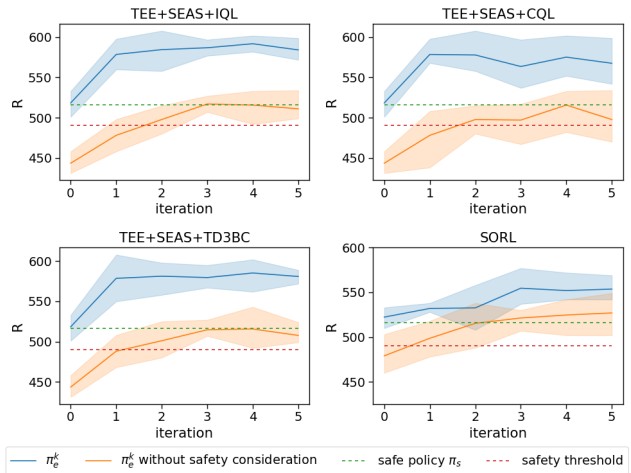

Figure 5: Safety constraint satisfaction.

- **IterIQL.** Iterative offline RL using IQL as the offline RL algorithm. No exploration noise is added, therefore $\pi_e^k = \pi^k$.
- **SORL [22].** SORL follows the iterative offline RL framework. The authors proposed V-CQL for offline policy training and designed an SER policy for safe and efficient exploration.

**Evaluation Metrics.** We evaluate the performance of a policy in terms of expected return. Besides, we check the safety of exploration policy by comparing average return in the collected dataset with the safety threshold $(1-\epsilon)J_s$.

Figure 4 presents the overall performance through iterations. Our proposed framework, combined with any of the three offline RL algorithms, substantially outperforms the baseline methods and achieves near-expert performance in approximately 5 iterations.

**Table 2: Performance of our proposed method in real-world experiments.**

| iteration | BuyCnt | ROI | CPA | ConBdg | GMV |
|---|---|---|---|---|---|
| 1 | +1.82% | +2.64% | -1.81% | -0.03% | +2.61% |
| 2 | +2.21% | +2.94% | -2.14% | +0.02% | +2.95% |
| 3 | +2.94% | +3.33% | -1.78% | +1.12% | +4.49% |
| 4 | +3.59% | +2.44% | -2.60% | +0.90% | +3.36% |

Both Iterative IQL+ASN and SORL get stuck in suboptimal policies after limited performance improvements, suffering from the performance bottleneck we discussed in section 4. Additionally, Iterative IQL without exploration achieves barely any performance improvement, which demonstrates the necessity of exploration in iterative offline RL.

We present the performance of exploration policies during the training process in Figure 5. Combined with any offline RL algorithm, our framework consistently ensures the performance of exploration policies to be above the safety threshold $(1 - \epsilon)J_s$, which validates the safety guarantee ability of SEAS. We also show the results when SEAS is omitted, in which the safety constraint is violated in early stages.

## 6.2 Online Experiments

We conduct real-world experiments on Alibaba display advertising platform. In each iteration, we utilize 20000 advertising campaigns to collect data for an epsiode, and employ IQL for offline training. Each episode contains 48 time steps, which means that the bidding parameter is adjusted every 30 minutes. After each policy update, we conduct a 10-day online A/B test using 1500 campaigns.

**Evaluation Metrics.** The return of the trained policy acts as our main metric of performance, and is referred to as BuyCnt in the online experiments. Additionally, we introduce several other metrics that are commonly used in the auto-bidding field.

- **BuyCut.** The total value of ad impressions won by the advertiser. Our objective $J$ in the RL formulation.
- **ROI.** The ratio between the total revenue and the consumed budget of the advertiser.
- **CPA.** Cost per aquisition, defined as the average cost for each successfully converted impression. A smaller CPA indicates a better performance of an auto-bidding policy.
- **ConBdg.** The total consumed budget of the advertiser.
- **GMV.** Gross Merchandise Volume, the total amount of sales over the campaign duration.

Table 2 shows the performance of our method in each iteration, compared with a static baseline policy trained by CQL[18] on a pre-collected dataset. We can see that the BuyCnt of our policy improves steadily through iterations, and our method consistently outperforms the baseline across all metrics.

## 6.3 Ablation Studies

We conduct ablation studies for a deeper analysis of the how different components work with each other in our method. Specifically, we aim to answer the following questions: (1) Do trajectory-wise exploration and trajectory-wise exploitation operate in close conjunction instead of being two independent components? (2) Does the reward model effectively reduce the influence of stochastic rewards in Robust Trajectory Weighting? (3) Does SEAS achieve the theoretical safety bound in practice? (4) While achieving the same safety bound, does SEAS sacrifices less performance than other baseline safety-guaranteeing methods?

To answer these questions, we conduct extensive experiments in the simulated environment described in Section 6.1.

**To answer Question 1.** We develop variants of TEE to delve deeper into how trajectory-wise exploration and trajectory-wise exploitation work together. We focus on one data-collection process followed by one offline training process. We omit SEAS in this experiment, to focus solely on TEE. An IQL policy is used as base policy $\pi$. Details of the variants are presented as follows:

- **TEE**. Given a policy $\pi$, we add PSN to construct $\pi_e$, then use $\pi_e$ to interact with the environment and collect dataset $D$. Then we train a new policy with IQL and Robust Trajectory Weighting on dataset $D$. The performance of the trained policy is presented.
- **w/o T-explore** removes trajectory-wise exploration (i.e. PSN). Instead, traditional ASN is used for $\pi_e$. To ensure a fair comparison, we control the strength of ASN to guarantee that the average return in $D$ are equal to that of PSN.
- **w/o T-exploit** removes trajectory-wise exploitation (i.e. Robust Trajectory Weighting). After collecting $D$ by PSN, we train a new policy with uniform sampling.
- **w/o TEE** removes TEE. ASN is used for exploration and uniform sampling for training.

Table 3 presents the performance of policies trained under different settings, and their performance gain over the base policy. We observe that TEE achieves significant performance improvement over the base policy under various budgets, while the absence of either component hurts its effectiveness. Interestingly, eliminating Trajectory-wise Exploitation leads to a substantial performance degradation. The reason behind this is that datasets collected by PSN contains a large fraction of undesirable actions, which are imitated by conservative algorithms. The above observation indicates that Trajectory-wise Exploration and Trajectory-wise Exploitation are inherently interconnected components, and their combination contributes substantially to the enhancement of the algorithm's performance.

**To answer Question 2.** We test the effectiveness of reward model in environments with different degrees of stochasticity. We first construct simulated environments with various stochasticity by controlling the variance of impression numbers per time step. In each environment, we collect an exploratory dataset with the same policy, and use two different data sampling strategies to train a policy on the dataset: a) Our proposed robust trajectory weighting. b) Trajectory weighting with the raw rewards $r_{i,t}$ instead of the reward model's prediction $\bar{r}_{i,t}$. We compare the performance of trained policies, to examine how reward model benefits the trajectory weighting method.

The results of the experiments are shown in Figure 6a, where $R_{robust}$ denotes the return of policy trained with Robust Trajectory Weighting, $R_{raw}$ represents the return of policy trained with raw

**Table 3: Ablation study on TEE. The last column presents the performance improvement over base policy. Best performance of each column is marked as bolded.**

| Ablation Settings | Budget | | | | | #Improve |
|---|---|---|---|---|---|---|
| | 1500 | 2000 | 2500 | 3000 | avg | |
| Base policy | 448.39 | 514.89 | 580.18 | 657.96 | 550.36 | - |
| TEE | **490.19±13.38** | **569.35±5.89** | **634.88±31.88** | 671.68 ± 59.69 | **591.53±24.79** | **+7.48%** |
| w/o T-explore | 431.09 ± 11.4 | 520.12 ± 12.05 | 594.14 ± 13.46 | 663.18 ± 14.97 | 552.13 ± 7.68 | +0.32% |
| w/o T-exploit | 365.53 ± 10.55 | 414.21 ± 29.05 | 495.89 ± 39.82 | 571.24 ± 51.27 | 461.72 ± 29.96 | -16.10% |
| w/o TEE | 414.38 ± 30.74 | 515.66 ± 10.35 | 607.47 ± 8.91 | **678.67±17.53** | 554.04 ± 7.13 | +0.66% |

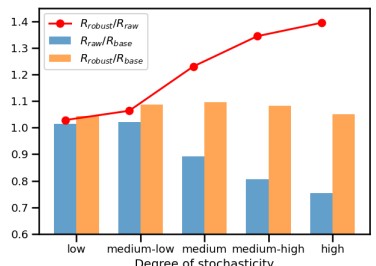

| Degree | # of Impr. |
|---|---|
| low | 175 |
| medium-low | $U_{[144,206]}$ |
| medium | $U_{[113,237]}$ |
| medium-high | $U_{[82,268]}$ |
| high | $U_{[50,300]}$ |

(b) Settings of different environments.

(a) Improvements of policy performance.

**Figure 6: The effectiveness of reward model in environments with different degrees of stochasticity.**

**Table 4: Safety-ensuring ability of SEAS.**

| $\epsilon$ | 0.4 | 0.3 | 0.2 | 0.1 | 0.05 | 0.01 |
|---|---|---|---|---|---|---|
| $1 - J(\pi_e^k)/J_s$ | 0.202 | 0.137 | 0.039 | -0.002 | -0.004 | -0.005 |

rewards, and $R_{base}$ the return of the data-collecting policy. Figure 6b shows the probability distributions of the impression number, where $U_{[a,b]}$ denotes a uniform distribution over $[a, b]$. The red line in Figure 6a indicates that the reward model is increasingly useful as the stochasticity of the environment intensifies. Moreover, the blue bars in the figure exhibit values lower than 1 in high stochasticity instances, which reflects that raw stochastic rewards could be misleading signals for trajectory weighting.

**To answer Question 3.** We fully evaluate the safety-ensuring ability of SEAS by setting different values of $\epsilon$ and observe the rate of performance decrease $1 - J(\pi_e^k)/J_s$, where $\pi_e^k$ is the policy produced by SEAS. We expect the safety constraint $1 - J(\pi_e^k)/J_s \leq \epsilon$ to be satisfied. In this experiment, we take a USCB [13] policy as the safe policy, and obtain its $Q$ function through fitting a dataset collected by itself.

From Table 4, we observe that SEAS consistently satisfies the safety constraint over a wide range of input $\epsilon$. Interestingly, for small $\epsilon$ values, the exploration policy generated by SEAS even outperforms the base policy.

**To answer Question 4.** We compare SEAS with two baseline safety-ensuring methods in terms of performance sacrificing, while

**Table 5: Different safety ensuring methods' impact on performance of trained policy. The third column presents the performance improvement of trained policy over base policy.**

| Safe Exploration Methods | Return | #Improve |
|---|---|---|
| No constraint | 594.25 ± 18.28 | +15.16% |
| SEAS | **572.35± 20.42** | **+10.92%** |
| Small noise | 528.07 ± 19.45 | +2.34% |
| Fixed range | 532.91 ± 6.74 | +3.28% |

ensuring safety to the same degree ($\epsilon = 0.05$). One USCB [13] policy is leveraged as the safe policy. In this experiment, we start from an IQL policy, preserve the design of TEE, and substitute SEAS with baselines presented below. Strength of PSN is $\sigma = 0.05$.

- **Small noise.** We limit the strength of PSN at a low level, by setting $\sigma = 0.01$.
- **Fixed range.** As proposed in [22], the safe exploratory action $a_e$ is given by $a_e \leftarrow clip(\pi_e(s_t), \pi_s(s_t) - \xi, \pi_s(s_t) + \xi)$. The safe policy $\pi_s$ is the same as that in SEAS, and $\xi$ is set to 0.1.

The results are presented in Table 5. We also show the result of not imposing any safety constraint, which is a performance upper bound. SEAS significantly outperforms the two baselines in terms of the average return of the trained policy. This observation suggests that the adaptive design of SEAS allows for minimal performance sacrifice.

## 7 CONCLUSION

This work presents a novel iterative RL framework for auto-bidding from a trajectory perspective, in order to boost the effectiveness of exploration and exploitation. We also pay particular attention on the safety of exploration in online advertisement systems and propose SEAS. Through comprehensive experiments, our method has been shown to be effective, achieving superior results compared to other baselines. In future work, we plan to test the effectiveness of TEE in other fields such as recommender systems and healthcare, where online policy training is challenging but urgently needed.

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

# A THEORETICAL JUSTIFICATION OF ROBUST TRAJECTORY WEIGHTING

We formally show that under the assumption of deterministic transition and stochastic reward, applying Robust Trajectory Weighting is equivalent to training offline RL on a dataset collected by a better behaviour policy.

For a dataset $D = \{\tau_i\}_{i=1}^{N}$ where $\tau_i = \{(s_{i,t}, a_{i,t}, s_{i,t+1}, r_{i,t})\}_{t=0}^{T}$. Each trajectory $\tau_i$ is collected by a different deterministic policy $\pi_i$, as in the case of Trajectory-wise Exploration. The behaviour policy $\pi$ of $D$ is then defined as sampling a policy from $\{\pi_i\}_{i=1}^{N}$ uniformly at the start of an episode, then acting according to the sampled policy till the end of the episode. Similarly, we define a weighted behaviour policy $\pi'$ as first sampling a policy from $\{\pi_i\}_{i=1}^{N}$ according to probabilities $\{w_i\}_{i=1}^{N}$, then acting with it. We aim to show that $J(\pi') \geq J(\pi)$.

The performance $J(\pi)$ could be expressed as expectation of value function over all possible initial states:

$$J(\pi) = \mathbb{E}_{s_0 \sim \rho}[V^{\pi}(s_0)], J(\pi') = \mathbb{E}_{s_0 \sim \rho}[V^{\pi'}(s_0)]. \tag{1}$$

For any initial state $s_0$, let $G_{s_0} = \{i|s_{i,0} = s_0\}$, $N_{s_0} = |G_{s_0}|$, we have

$$V^\pi(s_0) = \sum_{i \in G_{s_0}} V^{\pi_i}(s_{i,0})/N_{s_0}. \tag{2}$$

Similarly, for weighted behaviour policy $\pi'$,

$$V^{\pi'}(s_0) = \sum_{i \in G_{s_0}} w_i V^{\pi_i}(s_{i,0})/\sum_{i \in G_{s_0}} w_i. \tag{3}$$

Assuming deterministic transition and stochastic rewards of the underlying MDP, we have $V^{\pi_i}(s_{i,0}) = \bar{R}_i$, where $\bar{R}_i = \sum_{t=0}^T \gamma^t \bar{r}_{i,t}$ is the relabeled return in Robust Trajectory Weighting. Plugging it in (2) and (3) gives us

$$V^\pi(s_0) = \sum_{i \in G_{s_0}} \bar{R}_i/N_{s_0}, \tag{4}$$

$$V^{\pi'}(s_0) = \sum_{i \in G_{s_0}} w_i \bar{R}_i/\sum_{i \in G_{s_0}} w_i. \tag{5}$$

Subtracting (4) by (5),

$$V^{\pi'}(s_0) - V^\pi(s_0) \tag{6}$$

$$= \sum_{i \in G_{s_0}} w_i \bar{R}_i/\sum_{i \in G_{s_0}} w_i - \sum_{i \in G_{s_0}} \bar{R}_i/N_{s_0} \tag{7}$$

$$= \frac{1}{N_{s_0} \sum_{i \in G_{s_0}} w_i}(N_{s_0} \sum_{i \in G_{s_0}} w_i \bar{R}_i - (\sum_{i \in G_{s_0}} w_i)(\sum_{i \in G_{s_0}} \bar{R}_i)) \tag{8}$$

$$= \frac{1}{N_{s_0} \sum_{i \in G_{s_0}} w_i}(\sum_{i,j \in G_{s_0}, i<j}(w_i - w_j)(\bar{R}_i - \bar{R}_j)). \tag{9}$$

In (9), each term $(w_i - w_j)(\bar{R}_i - \bar{R}_j)$ inside the summation is non-negative, because

$$w_i = \exp((\bar{R}_i/V(s_{i,0}) - 1)/\alpha)/Z,$$

where $Z = \sum_{i=1}^N w_i$ is a normalization term and $\alpha \in \mathbb{R}^+$, is non-decreasing with respect to $\bar{R}_i$. Therefore, $V^{\pi'}(s_0) - V^\pi(s_0) \geq 0$ for every initial state $s_0$. Then applying (1) gives $J(\pi') \geq J(\pi)$.

The above derivation also highlights the significance of our proposed reward model. Relabeling rewards with the reward model's output makes $V^{\pi_i}(s_{i,0}) = \bar{R}_i$, allowing us to deal with stochastic rewards.

## B  PROOF OF THEOREM 1

THEOREM 1. *For any policy $\pi_e$ and any $\epsilon \in (0,1)$, given safe policies $\{\pi_s^i\}_{i=1}^n$ that satisfy $J(\pi_s^i) \geq J_s, \forall 1 \leq i \leq n$, the expected return $\mathbb{E}_\tau[R(\tau)]$ of trajectories generated by SEAS satisfies $\mathbb{E}_\tau[R(\tau)] \geq (1-\epsilon)J_s$.*

PROOF. In each time step, SEAS takes either action $a_e$ or $a_s$. For a trajectory $\tau$ generated by SEAS, let $t_0$ be the last step it takes exploratory action $a_e$. For those trajectories where it never takes $a_e$, set $t_0$ to 0. Let $temp_{t_0}$ denote the value of $temp$ at step $t_0$. Let $\mathbb{1}(\tau, s, a, i)$ be the indicator function of $s_{t_0}, a_{t_0}$ and $temp_{t_0}$ for trajectory $\tau$, defined as follows:

$$\mathbb{1}(\tau, s, a, i) = \begin{cases} \delta(s - s_{t_0}, a - a_{t_0}), & \text{if } i = temp_{t_0} \\ 0, & \text{otherwise,} \end{cases}$$

where $\delta(\cdot)$ is the Dirac delta function. Therefore

$$\forall \tau, \quad \sum_{i=1}^n \int_S \int_A \mathbb{1}(\tau, s, a, i)dads = 1. \tag{10}$$

Multiply both sides of (10) by $R(\tau)$,

$$\forall \tau, \quad \sum_{i=1}^n \int_S \int_A R(\tau)\mathbb{1}(\tau, s, a, i)dads = R(\tau).$$

Take expectation with respect to $\tau$ on both sides,

$$\mathbb{E}_\tau[R(\tau)] = \sum_{i=1}^n \int_S \int_A \mathbb{E}_\tau[R(\tau)\mathbb{1}(\tau, s, a, i)]dads.$$

Splitting trajectory $\tau$ by $t_0$, define $R(\tau^-) = \sum_{u=0}^{t_0-1} r_u$, $R(\tau^+) = \sum_{u=t_0}^T r_u$, then

$$\mathbb{E}_\tau[R(\tau)] = \sum_{i=1}^n \int_S \int_A \mathbb{E}_\tau[(R(\tau^-) + R(\tau^+))\mathbb{1}(\tau, s, a, i)]dads$$

$$= \sum_{i=1}^n \int_S \int_A \{\mathbb{E}_\tau[R(\tau^-)\mathbb{1}(\tau, s, a, i)]$$
$$+ \mathbb{E}_\tau[R(\tau^+)\mathbb{1}(\tau, s, a, i)]\}dads.$$

For trajectory $\tau$, since $t_0$ is the last time it takes $a_e$, all actions after $t_0$ follows safe policy $\pi_s^{temp_0}$. Therefore $E_\tau[R(\tau^+)\mathbb{1}(\tau, s, a, i)] = Q^{\pi_s^i}(s, a)\mathbb{1}(\tau, s, a, i)$, then

$$\mathbb{E}_\tau[R(\tau)] = \sum_{i=1}^n \int_S \int_A \{\mathbb{E}_\tau[R(\tau^-)\mathbb{1}(\tau, s, a, i)]$$
$$+ Q^{\pi_s^i}(s, a)\mathbb{1}(\tau, s, a, i)\}dads$$

$$= \sum_{i=1}^n \int_S \int_A \mathbb{E}_\tau[(R(\tau^-) + Q^{\pi_s^i}(s, a))\mathbb{1}(\tau, s, a, i)]dads$$

$$\geq \sum_{i=1}^n \int_S \int_A \mathbb{E}_\tau[(1-\epsilon)J_s\mathbb{1}(\tau, s, a, i)]dads$$

$$= \mathbb{E}_\tau[\sum_{i=1}^n \int_S \int_A (1-\epsilon)J_s\mathbb{1}(\tau, s, a, i)dads]$$

$$= \mathbb{E}_\tau[(1-\epsilon)J_s]$$

$$= (1-\epsilon)J_s,$$

where the inequality step is by line 6 in algorithm 1, and the following steps are from (10) and simple algebra. □

## C  DETAILS OF OFFLINE EXPERIMENTS

**Setup.** We construct a simulated advertising system for offline experiments. There are 30 advertisers competing for advertising impressions. An episode corresponds to one day in simulation, which is divided into 96 time steps. The number of impressions in each time step is random, and follows a uniform distribution on [50, 300]. The budget of each advertiser follows a uniform distribution on [1500, 3000]. Before an episode starts, the number of impressions in every time step, as well as the value of each impression for each advertiser is initialized. The state of an advertiser is three-dimensional: $[time, budget\_consumed, budget\_left]$. The reward of an advertiser is the total value she wins in all ad auctions during

one time step. Given current states and actions of all advertisers, the simulation returns next states and rewards. This is achieved by simulating ad auctions for all impressions in a single time step. For each impression, the system performs pre-ranking and ranking, and decides the winner of the auction. The winner gets the value of the impression, and pays for it according to the auction mechanism. We train a bidding policy for one advertiser while keeping other 29 advertisers' policy fixed. Therefore, the policies of other bidders could be seen as part of the environment which is stationary. The trained auto-bidding policy could serve for bidders with different budgets since the information of total budget is contained in the state representation. The implementation of the pre-ranking and ranking(auction) part follows from that in [22], where more details could be found.

**Parameter Settings.** Datasets collected in each iteration consists of 100000 transition tuples. Strength of trajectory weighting $\alpha$ is set to 0.1. Safety threshold $\epsilon$ is 0.05. PSN is implemented as factorised Gaussian noise [9] with $\sigma$ searched from [0.01,0.03,0.05] and kept fixed during training. ASN is Gaussian noise with $\sigma$ searched from [0.3,0.5,1]. In the IQL algorithm, the expectile parameter is set to 0.6 and $\beta$ is 1.25. Conservative factor is chosen as $\alpha = 0.8$ in CQL, and $\alpha = 2.5$ in TD3+BC. Implementation of SORL is consistent with the original paper [22] and code.

