# OpenReview forum: "Trajectory-wise Iterative Reinforcement Learning Framework for Auto-bidding"
_ACM.org/TheWebConf/2024/Conference — TheWebConf24 Oral_

### Official Review · Reviewer_wZZx · 2023-11-23

**Novelty:** 5
**Technical Quality:** 6

**Review:**

The paper addresses the problem of training auto-bidding tools for ad auctions. It studies the fact that current RL auto-bidding policies, trained offline on policies that are collected online, in an offline-online iterative framework; in this context, it identifies that the fact that offline training is too conservative. The paper hence proposes ways to enable more exploration in the offline training phase, while also controlling for a safe threshold for exploration.

Section 5 of the paper presents this approach, and Section 6 performs some synthetic experiments with other approaches. A real-world experiments is also presented, along with an ablation study.

The experimental section is quite well designed and seems convincing. So from a practical point of view it seems quite a good contribution.

The weakness is Section 6, however. It was not clear to me the reasoning behind the solutions to the two issues listed in Section 5.2. While the reasoning was given in the appendix, I think it should have been moved to the main text (in a shorter form), while adding some theoretical justification.

**Questions:**

1. Please address the question about the technical choices.

2. Do the authors intend to publish the code and data?

**Reviewer Confidence:**

3: The reviewer is confident but not certain that the evaluation is correct

**Scope:**

4: The work is relevant to the Web and to the track, and is of broad interest to the community

---

### Official Review · Reviewer_bo5A · 2023-11-28

**Novelty:** 5
**Technical Quality:** 4

**Review:**

The paper proposes Trajectory-wise Exploration and Exploitation (TEE), which introduces a novel data collecting and data utilization method for iterative offline RL from a trajectory perspective. To ensure the safety of online exploration while preserving the dataset quality for TEE, the paper proposes Safe Exploration by Adaptive Action Selection (SEAS).

pros:

(a) The paper studies an important real-world problem.

(b) The paper is well written and very easy to follow.

(c) The proposed approach is novel and has good intuitions.

(d) The paper has done comprehensive evaluations on a real-world platform. The paper also conducted extensive analysis (e.g., ablation study).

cons:

(a) The evaluated simulation environment (in Section 6.1) seems not to be a standard benchmark. There is only one simulation environment being evaluated. Is it possible to also compare the proposed approach to previous approaches, using some existing simulation environment?

(b) The paper may miss some recent works in the related work discussion.

Chen, Shuang, et al. "Model-Based Reinforcement Learning for Auto-bidding in Display Advertising." Proceedings of the 2023 International Conference on Autonomous Agents and Multiagent Systems. 2023.

Zhang, Haoqi, et al. "A Personalized Automated Bidding Framework for Fairness-aware Online Advertising." Proceedings of the 29th ACM SIGKDD Conference on Knowledge Discovery and Data Mining. 2023.

Korenkevych, Dmytro, et al. "Offline Reinforcement Learning for Optimizing Production Bidding Policies." arXiv preprint arXiv:2310.09426 (2023).

**Questions:**

The evaluated simulation environment (in Section 6.1) seems not to be a standard benchmark. There is only one simulation environment being evaluated. Is it possible to also compare the proposed approach to previous approaches, using some existing simulation environment?

**Reviewer Confidence:**

2: The reviewer is willing to defend the evaluation, but it is likely that the reviewer did not understand parts of the paper

**Scope:**

4: The work is relevant to the Web and to the track, and is of broad interest to the community

---

### Official Review · Reviewer_EQRw · 2023-12-14

**Novelty:** 6
**Technical Quality:** 7

**Review:**

The paper presents an iterative RL framework that utilizes multiple auto-bidding agents to tackle issues in online advertising, as RL algorithms for auto-bidding encounter reduced performance due to safety considerations. The authors introduce Trajectory-wise Exploration and Exploitation, incorporating Parameter Space Noise for exploration and a trajectory weighting algorithm for exploitation. Additionally, they introduce SEAS to guarantee the safety of online exploration. Experiments conducted on Alibaba’s display advertising platform validate the effectiveness of the proposed approach.


Pros
The design of TEE and PSN is quite novel and can help address the challenges of exploration. The resultant datasets will have more dispersed trajectory return distributions making exploration more effective.
The proposed trajectory weighting algorithm helps overcome the conservatism problem in offline RL and results in high quality trajectories.
The authors tap into a very important concern in RL systems- “Safety in Exploration” and propose SEAS to  determine safe exploration actions.
Cons
The suggested method appears to be more intricate and involving multiple iterations compared to current approaches. Implementing it in real-world scenarios could pose challenges.
While the authors recognize the decline in performance in simulated settings- their proposed remedy continues to depend on an iterative offline RL framework, potentially not completely eliminating the reliance on simulations.

**Questions:**

What is the computational overhead associated with the proposed methods and how scalable are they compared to existing approaches. A latency analysis would be helpful.

**Reviewer Confidence:**

3: The reviewer is confident but not certain that the evaluation is correct

**Scope:**

4: The work is relevant to the Web and to the track, and is of broad interest to the community

---

### Official Review · Reviewer_EEpr · 2023-12-16

**Novelty:** 5
**Technical Quality:** 3

**Review:**

This paper presents a reinforcement learning approach for pacing in the the context of autobidding in the context of advertising in demand side platforms. The paper's key contributions are as follows: (a) a method for trajectory wise exploration and exploitation (TEE) for combating the overly conservative behavior when using off-the-shelf offline RL algorithms, (b) a safe exploration method (SEAS) when using exploratory policies for data collection. Through an repeated application of safe data collection with exploratory policies and use of offline RL methods with TEE, the paper claims that they achieve near optimal behavior of the final resulting policy for pacing.

The paper is well motivated and addresses a very interesting and technically challenging question with practical importance. These components (described above) are not particularly novel within the reinforcement learning literature, although, their applications certainly appears novel to my knowledge.

**Questions:**

I will present a few more details about each of these components and some remarks surrounding these choices:

1. Perhaps I might have missed this, but, any of the results in the paper (aside from online experiments) have returns marked on them, and I wasn't sure how these are measured since counterfactuals (what happened if the agent bid differently at anytime) is a very non-trivial quantity to estimate; for example, this might depend on quantities such as conversions which are inherently hard to estimate and/or fit using a machine learning model. I can imagine two different ways of doing this: simulator (model based) - this carries the risk of the estimates being biased due to sim2real issues, and this is precisely what the paper aims to move away from; using replay buffer (model free) - this needs some form of importance weighting or other off-policy evaluation methods such as importance sampling which have variance related issues. It appears fairly cumbersome to try out these ablations on an online experiment since it will ruin value to advertiser.

2. The first "E" in the TEE represents trajectory wise exploration. The paper achieves this through Parameter Space Noise (PSN) as opposed to the more well known Action Space Noise (ASN). This is an interesting idea - particularly since they coincide for linear policies. The variance of the noise seems to be a particularly hard hyper-parameter to work with especially as the parameters of the policy change - so perhaps some more clarification on the sensitivity of this parameter (as policy parameters change) can be useful to discuss.

3. The second "E" in TEE is the trajectory wise exploitation, where, trajectories with higher returns are emphasized during training; recognizing the issue that the returns are highly stochastic and have outlier issues, the authors propose to use fitted model to estimate returns, and use least squares. The question here is that isn't least squares also a very non-robust loss function when dealing with outliers? Perhaps the more suitable alternative here would be to use robust variants such as Huber loss for the function fitting as there is no reason why the least squares fit would in anyway help mitigate the heavy tailed nature of the collected data/targets, even on an average case.

4. Getting to the safe exploration component (SEAS): This seems like a very nice idea, although might need more connections with the safe Reinforcement Learning literature (this is a minor comment, can be addressed). However, in terms of practicality, again, during inference time, this requires the Q-function (which measures expected long term return from the current state) for each of the safe policies. While in theory, it does make sense to rule out actions based on the Q-value, getting an estimate of this is non-trivial and unclear to me. If this is a fitted function, this requires more floating point operations/inference cost that may or may not meet the bar for deployment, and its quality may not even be completely worth relying on.

All in all, while the paper presents an interesting technical framework for a highly relevant practical problem, I am somewhat left unconvinced by the detail/practicality surrounding various components of the proposed framework.

**Reviewer Confidence:**

3: The reviewer is confident but not certain that the evaluation is correct

**Scope:**

4: The work is relevant to the Web and to the track, and is of broad interest to the community

---

### Decision · Program_Chairs · 2024-01-22

**Decision:**

Accept (Oral)

**Comment:**

This is the meta-review by the SPC responsible for your paper, and takes into account the opinions expressed by the referees, the subsequent decision thread, and my own opinions about your work.

 - This paper proposes Trajectory-wise Exploration and Exploitation, which introduces a data collecting and data utilization method for iterative offline RL from a trajectory perspective. In addition, the authors proposed Safe Exploration by Adaptive Action Selection to ensure the safety of online exploration. Both offline and real-world experiments were conducted.
 - the proposed approach is quite novel and has good intuitions, and the design of the experimental section is comprehensive and convincing.